# A multivariate genome-wide association study of psycho-cardiometabolic multimorbidity

**Vilte Baltramonaityte** [1], **Jean-Baptiste Pingault**[2,3], **Charlotte A. M. Cecil**[3,4,5], **Priyanka Choudhary**[6], **Marjo-Riitta Järvelin**[6,7], **Brenda W. J. H. Penninx**[8], **Janine Felix**[9,10], **Sylvain Sebert**[6], **Yuri Milaneschi**[8], **Esther Walton** [1]*, on behalf of the EarlyCause Consortium¶

**1** Department of Psychology, University of Bath, Bath, United Kingdom, **2** Department of Clinical, Educational, and Health Psychology, University College London, London, United Kingdom, **3** Department of Child and Adolescent Psychiatry/Psychology, Erasmus MC, University Medical Center Rotterdam, Rotterdam, the Netherlands, **4** Department of Epidemiology, Erasmus MC, University Medical Center Rotterdam, Rotterdam, the Netherlands, **5** Molecular Epidemiology, Department of Biomedical Data Sciences, Leiden University Medical Center, Leiden, the Netherlands, **6** Research Unit of Population Health, University of Oulu, Oulu, Finland, **7** Department of Epidemiology and Biostatistics, School of Public Health, Imperial College London, London, United Kingdom, **8** Department of Psychiatry, Amsterdam Public Health and Amsterdam Neuroscience, Amsterdam UMC, Vrije Universiteit, Amsterdam, the Netherlands, **9** The Generation R Study Group, Erasmus MC, University Medical Center Rotterdam, Rotterdam, the Netherlands, **10** Department of Pediatrics, Erasmus MC, University Medical Center Rotterdam, Rotterdam, the Netherlands

¶ Membership of the EarlyCause Consortium is available here https://earlycause.europescience.eu/partners.
* E.Walton@bath.ac.uk

**Data Availability Statement:** To access summary statistics for psycho-cardiometabolic multimorbidity, a data transfer agreement is required with 23andMe (dataset-

## Abstract

Coronary artery disease (CAD), type 2 diabetes (T2D) and depression are among the leading causes of chronic morbidity and mortality worldwide. Epidemiological studies indicate a substantial degree of multimorbidity, which may be explained by shared genetic influences. However, research exploring the presence of pleiotropic variants and genes common to CAD, T2D *and* depression is lacking. The present study aimed to identify genetic variants with effects on cross-trait liability to psycho-cardiometabolic diseases. We used genomic structural equation modelling to perform a multivariate genome-wide association study of multimorbidity ($N_{effective}$ = 562,507), using summary statistics from univariate genome-wide association studies for CAD, T2D and major depression. CAD was moderately genetically correlated with T2D ($r_g$ = 0.39, $P$ = 2e-34) and weakly correlated with depression ($r_g$ = 0.13, $P$ = 3e-6). Depression was weakly correlated with T2D ($r_g$ = 0.15, $P$ = 4e-15). The latent multimorbidity factor explained the largest proportion of variance in T2D (45%), followed by CAD (35%) and depression (5%). We identified 11 independent SNPs associated with multimorbidity and 18 putative multimorbidity-associated genes. We observed enrichment in immune and inflammatory pathways. A greater polygenic risk score for multimorbidity in the UK Biobank ($N$ = 306,734) was associated with the co-occurrence of CAD, T2D and depression (OR per standard deviation = 1.91, 95% CI = 1.74–2.10, relative to the healthy group), validating this latent multimorbidity factor. Mendelian randomization analyses suggested potentially causal effects of BMI, body fat percentage, LDL cholesterol, total cholesterol, fasting insulin, income, insomnia,

request@23andMe.com) before making a request to the University of Bath Research Data Archive (https://doi.org/10.15125/BATH-01179). Further information regarding access to 23andMe is available at: https://research.23andme.com/collaborate/. Summary statistics for the top 10,000 SNPs generated during this study are available from the University of Bath Research Data Archive: https://doi.org/10.15125/BATH-01179. Summary statistics for coronary artery disease can be obtained from: http://www.cardiogramplusc4d.org. Summary statistics for type 2 diabetes can be obtained from: http://diagram-consortium.org/downloads.html. To access summary statistics for depression, a data transfer agreement is required from 23andMe (dataset-request@23andMe.com) before a request is made to David Howard (D. Howard@ed.ac.uk), as described in: https://www.nature.com/articles/s41593-018-0326-7. UK Biobank data can be accessed via an application process outlined here: https://www.ukbiobank.ac.uk/enable-your-research/apply-for-access. Code underlying our analyses can be found on GitHub: https://github.com/VilteBaltra/Psycho-cardiometabolic-multimorbidity.

**Funding:** This project has received funding from the European Union's Horizon 2020 research and innovation programme under Grant Agreement No 848158 (EarlyCause). The funders had no role in study design, data collection and analysis, decision to publish, or preparation of the manuscript.

**Competing interests:** The authors declare no competing interests.

and childhood maltreatment. These findings advance our understanding of multimorbidity suggesting common genetic pathways.

## Author summary

While observational research has shown a substantial degree of overlap between depression, coronary artery disease and type 2 diabetes, few studies have attempted to identify genetic variants associated with multimorbidity between these conditions. Here, we explore the shared genetic architecture of depression, coronary artery disease and type 2 diabetes (i.e., psycho-cardiometabolic diseases) and examine common genetic variants associated with the co-occurrence of these conditions. Employing a novel method for performing multivariate genome-wide association studies, we show that there are 11 independent genetic variants across nine distinct genomic risk loci associated with psycho-cardiometabolic multimorbidity. We observe enrichment in immune and inflammation-related pathways and identify 18 multimorbidity-associated genes. We show that the polygenic risk score developed based on our multimorbidity genome-wide association study is predictive of the co-occurrence of depression, coronary artery disease and type 2 diabetes in an independent sample. Lastly, we identify eight potentially causal risk factors for multimorbidity. These results advance our understanding of the shared genetic influences in psycho-cardiometabolic diseases.

## Introduction

Depression, coronary artery disease (CAD) and type 2 diabetes (T2D) are important public health issues. Whilst each of these chronic disorders alone represent a major global burden, multimorbidity between them presents an additional challenge for healthcare systems [1–3]. Epidemiological studies suggest that individuals with depression have an 80–90% greater risk of cardiovascular morbidity and mortality [4] and a 32–60% higher risk of T2D [5,6] than individuals without depression. The reverse association has also been observed, with approximately 40% of people with CAD and 18–28% of people with diabetes either meeting the criteria for depression or experiencing depressive symptoms [7,8]. Notably, life expectancy in individuals with a diagnosis of depression is reduced [9], which may be partially accounted for by the co-occurrence with physical health diseases [10,11]. This emphasizes the importance of understanding the mechanisms through which mental and physical diseases may co-occur.

The relationship between depression, CAD and T2D may be attributed in part to shared lifestyle and other risk factors such as lack of physical activity [12,13], unhealthy diet [14,15], increased body mass index (BMI) [16–18], altered hypothalamic-pituitary-adrenal axis [19,20], inflammation [21–23] and childhood trauma [24]. For instance, BMI and inflammation have been causally linked to all three disorders in Mendelian randomization studies [25–30], albeit with some conflicting findings [31–33]. However, in several meta-analyses, estimates of the CAD-depression and T2D-depression relationship were similar before and after adjustment for major sociodemographic and lifestyle indicators [5,34–36], suggesting that these indicators do not entirely explain the association. Another plausible explanation for multimorbidity between these conditions is the presence of shared genetic aetiology (i.e., pleiotropic genes) that function as a hub linking these disorders [2]. In line with this, twin and family studies reveal moderate genetic correlations between depression and CAD (42%) [37], and depression and T2D (up to 25%) [38]. However, recent studies based on genome-wide

association data and polygenic risk scores provide conflicting evidence [39–41], with only some studies observing significant positive genetic correlations among the above mentioned traits [42–44]. The genetic overlap between CAD and T2D was more consistent, with multiple studies reporting a significant positive genetic correlation [42,45,46].

Despite many studies investigating the genetic overlap between depression, cardiovascular and metabolic diseases [e.g., 47], the latent genetic factor structure across all three diseases has not been explored. Additionally, research exploring the presence of pleiotropic variants and genes that are common to depression, CAD, *and* T2D is lacking. Characterizing multivariate genetic associations with psycho-cardiometabolic diseases (where *psycho* stands for depression, *cardio* for CAD and *metabolic* for T2D) and understanding the biological mechanisms that contribute to multimorbidity between these conditions is important. It would allow us to examine causal risk factors for multimorbidity (e.g., by providing genetic instruments for Mendelian randomization analysis) and help to identify potentially engageable treatment targets.

Accordingly, the aims of this study were to: (1) model the shared genetic architecture of depression, CAD and T2D with a latent multimorbidity factor; (2) identify genetic variants associated with multimorbidity; (3) perform functional gene mapping to determine if the prioritised genes are enriched in specific tissues or biological pathways; and (4) validate a polygenic risk score for multimorbidity within an independent sample.

## Methods

### Ethics statement

This research was conducted using the UK Biobank resource, application number 65769. The UK Biobank study was conducted under generic approval from the National Health Service (NHS) Research Ethics Service. The study protocol used by 23andMe was approved by an external Association for Accreditation of Human Research Protection Programs (AAHRPP)-accredited institutional review board. All cohorts contributing to the present study obtained written informed consent from all participants. Additionally, ethical approval for the present study was obtained from the University of Bath (PREC: 20–195).

### GWAS selection

First, we identified the largest univariate genome-wide association meta-analyses available to date from individuals of predominantly European ancestry for three distinct phenotypes: major depression [44], CAD [48], and T2D [42] (Table 1; S1 Appendix methods section). We avoided using genome-wide association studies (GWASs) with mixed ancestry groups (i.e., > 25% non-European ancestry individuals), as they may bias results from genetic factor analysis [49]. As a second step, we selected GWASs that do not include the UK Biobank (UKBB) cohort (i.e., Scott et al. [50] for T2D and Howard et al. [44] for depression with UKBB removed), as we planned to use this cohort as an independent replication sample for polygenic risk score (PRS) analysis. For detailed characteristics of the input populations and the sources of data see S1 Table. For a flowchart of all our analyses see Fig A in S1 Appendix.

### Single-trait heritability, genetic correlations, and factor analysis

We used linkage disequilibrium (LD) score regression within Genomic structural equation modelling (Genomic SEM, version 0.0.5) [49] to estimate the heritability of depression, CAD and T2D, as well as the genetic correlations among the traits. For quality control steps, see S1 Appendix methods section. Subsequently, Genomic SEM was used to perform a genetic

**Table 1. A list of Contributing Genome-Wide Association Studies.**

| Phenotype | Used in | GWAS | Year | Cases | Controls |
|---|---|---|---|---|---|
| **CAD** | Discovery GWAS for Genomic SEM | Nikpay et al. [48] | 2015 | 60,801 | 123,504 |
| | Discovery GWAS for PRS | Nikpay et al. [48] | 2015 | 60,801 | 123,504 |
| **T2D** | Discovery GWAS for Genomic SEM | Mahajan et al. [42] | 2018 | 74,124 | 824,006 |
| | Discovery GWAS for PRS | Scott et al. [50] | 2017 | 26,676 | 132,532 |
| **MD** | Discovery GWAS for Genomic SEM | Howard et al. [44] | 2019 | 246,363 | 561,190 |
| | Discovery GWAS for PRS | Howard et al. [44] (no UKBB) | 2019 | 140,045 | 378,325 |

Univariate GWASs contributing to the multivariate GWAS of psycho-cardiometabolic multimorbidity. All summary statistics are based on individuals of European ancestry, apart from Nikpay et al. [48], which also includes individuals from mixed ancestry groups (with 77% being European). Summary statistics used to construct the PRS exclude the UK Biobank cohort. GWAS, genome-wide association study; PRS, polygenic risk score; UKBB, UK Biobank.

factor analysis using diagonally weighted least squares estimation. To estimate the shared variance between depression, CAD, and T2D, a common factor model was specified with all three traits loading onto a latent *multimorbidity* factor.

## Multivariate GWAS and heterogeneity index

We used Genomic SEM to carry out a multivariate GWAS whereby the latent factor of multimorbidity (obtained in the previous step) was regressed on each SNP. This permitted a new set of summary statistics to be estimated for this common factor. SNP effects for the latent factor were estimated only for SNPs which were present in each of the univariate summary statistics files, resulting in 6,820,149 SNPs. A follow-up model was specified to obtain a heterogeneity Q index for each SNP. This index indicates the extent to which the SNP effect deviates from the common factor structure, with larger $Q_{SNP}$ values suggesting greater heterogeneity. Heterogeneous SNPs are unlikely to affect all phenotypes via the common factor but, instead, are more likely to be phenotype-specific, and are thus not indicative of multimorbidity. SNPs with Q estimates significant at the genome-wide level ($Q_{SNP} P < 5e-8$) and with directionally discordant univariate effect estimates were interpreted as potentially heterogeneous. For details, see methods section in S1 Appendix.

## Functional annotation and gene mapping

Functional mapping and annotation of genetic associations was performed using FUMA GWAS online platform [51] version 1.5.2d. We used the SNP2GENE pipeline with default settings to identify independent genome-wide significant SNPs ($P < 5e-8$) in low LD ($r^2 < 0.1$). LD blocks of independent significant SNPs that are located next to each other ($< 250$ kb apart) were merged into one genomic risk locus. To ensure that functional analysis captures the multimorbidity signal and is not driven by any single disease, we removed all SNPs with evidence for heterogenous effects ($Q_{SNP} P < 5e-8$ and directionally discordant univariate effect estimates) prior to annotation. Given that independent SNPs might not be causal themselves, but instead in close proximity of causal SNPs, we broadened the genomic loci for annotation to include all known variants that are available in the 1000G reference panel and are in LD ($r^2 \geq 0.6$) with one of the non-heterogeneous, independent significant SNPs, as done elsewhere [51] (methods section in S1 Appendix).

Subsequently, to understand which genes may be involved in multimorbidity, functionally annotated SNPs were mapped to genes based on positional mapping, expression quantitative trait loci (eQTL) and chromatin interactions (S1 Appendix methods section). Default

parameters were selected for each of these analyses (S1 Text). LocusZoom platform [52] was used to obtain regional visualization plots of key genomic risk loci.

### Pathway enrichment analysis and tissue specificity

All genes identified via at least one of these mapping techniques were used as input in FUMA's GENE2FUNC pipeline, which annotated the prioritised genes in a biological context (S1 Appendix methods section; S2 Text). Enrichment of input genes was tested using curated gene sets and GO terms obtained from MSigDB [53] database and WikiPathways [54], whereas tissue specificity analysis was performed in 54 specific and 30 general tissue types based on GTEx v8 data [55].

### Gene-based and gene-set analyses

Given that the effects of individual SNPs can be too weak to detect when dealing with polygenic traits, MAGMA [56] gene-based and gene-set analyses were performed (as implemented within the FUMA platform [51]) to determine the joint effect of multiple SNPs within a given gene. For gene-based analysis, the degree of association for gene with multimorbidity was quantified using gene-based $P$-values, which were obtained by assigning input SNPs to genes when these were physically located within the gene or within 10kb window on either side. For gene-set analysis, gene-set $P$-values were computed for curated gene sets and GO terms obtained from the MsigDB[53] database. Unlike pathway enrichment analysis implemented within the GENE2FUNC pipeline, which only tests for enrichment of prioritized genes, MAGMA gene-set analysis was performed using the full distribution of genetic associations [51]. Significance for gene set analysis was defined as alpha divided by the total number of protein coding genes tested.

### Polygenic risk score analysis

To evaluate how well our GWAS for multimorbidity captures multimorbidity risk, we performed polygenic risk score (PRS) analysis. To do this, we first repeated the multivariate GWAS using summary statistics of European ancestry that do not include the UKBB cohort (Table 1). Subsequently, a PRS was calculated using PRSice-2 [57] to assess its association with phenotypic multimorbidity. Summary statistics for multimorbidity (excluding UKBB) provided the allelic weightings for each SNP, which were used to generate polygenic risk scores for 306,734 individuals in the UKBB cohort–our independent target sample–adjusting for 10 genetic principal components (PCs), sex and age. Multimorbidity was defined as an ordinal variable, where 0 = no disease, 1 = any one disease, 2 = any two diseases, and 3 = all three diseases (i.e., depression, CAD and T2D). See methods section in S1 Appendix for further details. Subsequently, a multinomial logistic regression controlling for sex and age was performed to investigate the degree to which the PRS was associated with multimorbidity. To ensure that SNPs with higher heterogeneity estimates do not confer disproportionate liability to any individual trait, we repeated the PRS analysis with non-heterogeneous SNPs only ($Q_{SNP}$ $P \geq$ 5e−8).

To compare how well a multimorbidity-based PRS performs in comparison to PRSs based on single diseases, we generated PRSs for the individual phenotypes as well. Significant differences in the four PRSs (i.e., multimorbidity, CAD, T2D, depression) were tested using one-way ANOVA with post-hoc Tukey HSD to account for multiple comparisons.

### Genetic correlations and Mendelian randomization

As a last step, we used the LD score regression tool v1.0.1 [45,58] implemented in Python v2.7.18 to estimate genetic correlations between multimorbidity and 18 risk factors, such as BMI, blood pressure, cholesterol, inflammation, neuroticism, childhood maltreatment and income (see S1 Appendix methods section for a complete list). For comparability, genetic correlations between the three contributing GWASs (CAD, T2D, depression) and the risk factors were also obtained. Bonferroni correction was applied to account for multiple comparisons.

To further explore which risk factors may be causal for multimorbidity, we performed inverse-variance weighted (IVW) two-sample Mendelian randomization (MR) analysis between selected risk factors and multimorbidity using the TwoSampleMR package [59]. To ensure our outcome GWAS captures multimorbidity, we removed SNPs with evidence of heterogeneity ($Q_{SNP}$ $P$ < 5e−8 and directionally discordant effect estimates). A series of sensitivity analyses were also carried out (S1 Appendix, methods section). All MR analyses were conducted using multimorbidity GWAS without the UKBB. This helped to reduce bias in MR estimates due to sample overlap as eight risk factors were solely based on the UKBB. To account for non-UKBB related sample overlap, we obtained bias-corrected IVW-MR estimates using the recently developed MRlap package[60]. MRlap adjusts for biases due to overlapping samples, weak instruments, and winner's curse by incorporating cross-trait LD-score regression to approximate sample overlap [60].

## Results

### Heritability and genetic correlations

Heritability estimates (reported on the liability scale) were similar across all three univariate traits: 0.070 (*SE* = 0.005) for CAD, 0.162 (*SE* = 0.008) for T2D, and 0.064 (*SE* = 0.002) for depression. LD score regression identified a significant moderate correlation between CAD and T2D ($r_g$ = 0.39, *SE* = 0.03, *P* = 2e-34), and significant but weak correlations between CAD and depression ($r_g$ = 0.13, *SE* = 0.03, *P* = 3e-6), and between depression and T2D ($r_g$ = 0.15, *SE* = 0.02, *P* = 4e-15). For heritability Z scores, see S2 Table.

### Factor analysis

We specified a common factor model where all three traits loaded onto the same multimorbidity factor. T2D loaded most highly on this factor, followed by CAD and depression (Fig 1). Accordingly, the common factor explained the largest proportion of variance in T2D ($R^2$ = 0.45), followed by CAD ($R^2$ = 0.35) and depression ($R^2$ = 0.05). Hence, the common factor explained on average 28.3% of the total standardised genetic variance between the three traits. Model fit indices were not available due to specifying a fully saturated model (*df* = 0).

### Multivariate GWAS and heterogeneity index

We identified 389 SNPs associated with multimorbidity, of which 11 were independent (Table 2, Figs 2 and B in S1 Appendix). The independent SNPs were distributed across nine genomic loci (S3 Table). The $Q_{SNP}$ heterogeneity estimate was significant for six of the 11 SNPs (*P* < 5e-8; S4 Table; Fig C in S1 Appendix).

An inspection of the univariate betas within the three contributing GWASs revealed directionally discordant estimates across the three traits for four of the 11 independent SNPs (S5 Table), indicating that these particular SNPs were unlikely to operate via the common factor but, instead, were more likely to be phenotype specific. The remaining seven SNPs were consistent in the direction of their univariate betas, with four SNPs showing the largest effects

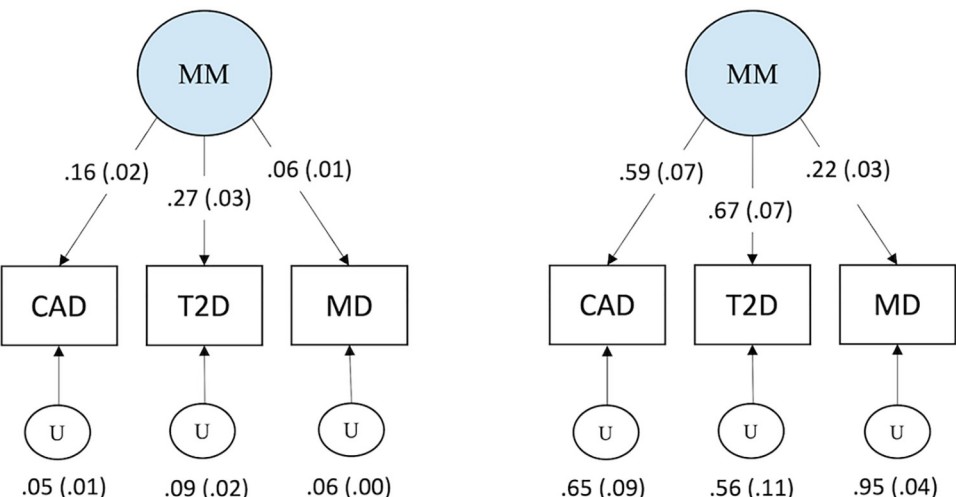

**Fig 1. A Common Factor Model for Psycho-Cardiometabolic Multimorbidity.** Unstandardized coefficients (SE) on the left and standardized coefficients (SE) on the right for the genetically defined common factor of multimorbidity. The model uses unit variance identification for the latent factor. All paths are significant at $P < 2e\text{-}13$. MM, multimorbidity; CAD, coronary artery disease; T2D, type 2 diabetes; MD, major depression; U, residual variance.

for CAD and T2D, and three SNPs having comparable estimates across the three traits (for regional plots of these SNPs see Fig 2). While we observed moderate genomic inflation (i.e., $\lambda_{GC}$ of 1.69), an intercept of 0.99 ($SE = 0.01$) indicated that it was likely due to polygenicity rather than uncontrolled inflation (S2 Table). The effective sample size for the multimorbidity GWAS was 562,507.

**Table 2. Independent Significant SNPs at $r^2 < 0.1$ identified in the Multivariate GWAS of Psycho-Cardiometabolic Multimorbidity.**

| Locus | rsID | Chr | Position | *P*-value | LD SNPs | GWAS SNPs | $Q_{SNP}$ *P*-value | Nearest Gene |
|---|---|---|---|---|---|---|---|---|
| 1 | rs10789340 | 1 | 72940273 | 3.38E-10 | 580 | 394 | 3.15e-12 | *RPL31P12* |
| 2 | rs9349379* | 6 | 12903957 | 3.17E-19 | 222 | 135 | 2.91e-27 | *PHACTR1* |
| 3 | rs10455872* | 6 | 161010118 | 4.81E-15 | 200 | 143 | 1.55e-25 | *LPA* |
| 3 | rs186696265* | 6 | 161111700 | 3.78E-11 | 91 | 56 | 7.53e-21 | *RP1-81D8.3* |
| 4 | rs2043539 | 7 | 12253880 | 1.23E-08 | 329 | 210 | 3.52e-07 | *TMEM106B* |
| 5 | rs3731239 | 9 | 21974218 | 3.03E-09 | 167 | 113 | 3.45e-06 | *RP11-145E5.5: CDKN2A* |
| 5 | rs2891168 | 9 | 22098619 | 6.88E-74 | 215 | 155 | 1.36e-27 | *CDKN2B-AS1* |
| 6 | rs532436 | 9 | 136149830 | 2.88E-08 | 168 | 65 | 8.93e-06 | *ABO* |
| 7 | rs34872471* | 10 | 114754071 | 2.32E-11 | 133 | 109 | 1.85e-17 | *TCF7L2* |
| 8 | rs2004910 | 12 | 121374727 | 3.60E-09 | 493 | 337 | 0.003 | *RPL12P33* |
| 9 | rs1962412 | 17 | 46970259 | 2.57E-08 | 510 | 359 | 0.095 | *SUMO2P17: ATP5G1* |

rsID, unique identifier of independent significant single nucleotide polymorphisms (SNPs); Chr, chromosome; Position, position on hg19. LD SNPs = the number of SNPs in linkage disequilibrium (LD) with the corresponding independent significant SNP. This includes non-GWAS-tagged SNPs extracted from 1000G reference panel. GWAS SNPs = number of multimorbidity GWAS-tagged SNPs in LD ($r^2 < 0.1$) with the corresponding independent significant SNP filtered by $P \leq 0.05$. $Q_{SNP}$ *P*-value = test for violation of the null hypothesis that the SNP acts entirely through the common factor. An asterisk indicates heterogeneous SNPs with a $Q_{SNP}$ $P < 5e\text{-}8$ and directionally discordant univariate betas. While another two SNPs had $Q_{SNP}$ $P < 5e\text{-}8$ indicative of heterogeneity, their univariate beta estimates were directionally concordant.

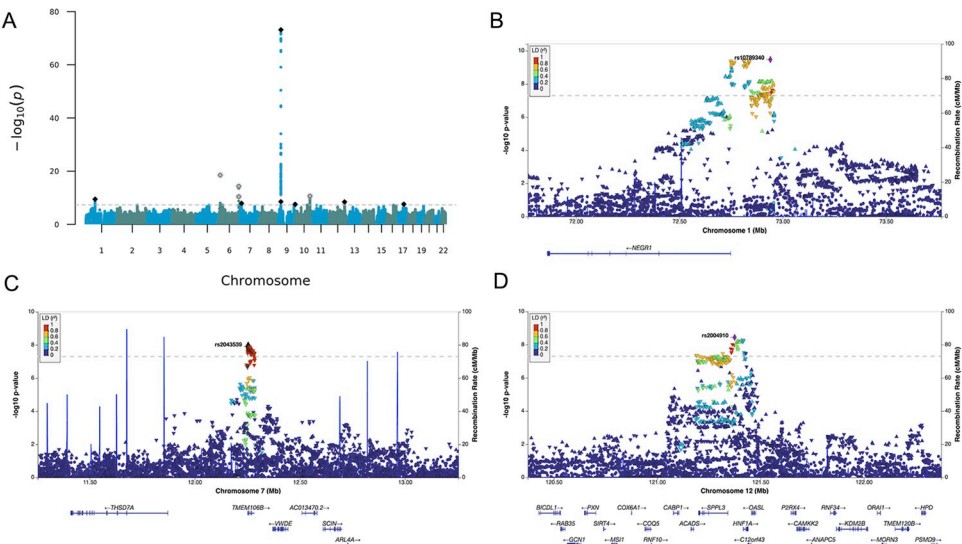

**Fig 2. Manhattan and LocusZoom Plots of the Multivariate GWAS of Psycho-Cardiometabolic Multimorbidity.**
(A) A Manhattan plot displaying the results for the multivariate GWAS of psycho-cardiometabolic multimorbidity obtained using Genomic SEM (with Diagonally Weighted Least Squares estimation). The y axis depicts $-\log10(P)$ values for variants associated with multimorbidity. The dashed, horizontal grey line denotes the genome-wide significance threshold at $P$ = 5e-8. Points above the grey line represent genome-wide significant hits. The black diamonds represent independent hits. The grey stars represent independent SNPs with evidence for heterogeneous effects ($Q_{SNP}$ $P < 5e-8$ and directionally discordant univariate effect estimates). (B, C, D) Regional plots centered on three top variants (rs10789340, rs2043539 and rs2004910, respectively) that have comparable univariate estimates across coronary artery disease, type 2 diabetes and depression. Coding genes are shown in the panel below. The blue line represents the recombination rate.

## Functional annotation and gene mapping

To ensure that functional analysis captured multimorbidity and was not driven by any single disease, we only included non-heterogeneous SNPs and those in LD. This left 1,562 GWAS tagged SNPs, which were functionally annotated to genes based on positional mapping, eQTL associations and chromatin interactions using FUMA. A total of 200 unique genes were implicated by at least one of these mapping techniques while 37 were identified using all three methods. For a complete list of prioritised genes refer to S6 Table.

## Gene-based analysis of all SNPs

Gene-based analysis using MAGMA[56] provided *P*-values for the joint association effect of *all* non-heterogeneous SNPs. Non-heterogeneous SNPs were mapped to 18,931 protein coding genes, with 122 of these genes being identified as significant after correcting for the number of genes tested ($P$ = 0.05/18931 = 2.64e-6). Eighteen genes identified using MAGMA overlapped with the genes implicated by FUMA, providing stronger support for the involvement of these particular genes (Table 3).

## Pathway enrichment analysis and tissue specificity of prioritised genes

The prioritised genes demonstrated enrichment in 10 Reactome pathways, three GO molecular functions, 43 GO biological processes, nine KEGG and 25 canonical pathways, among others. Based on an FDR-adjusted *P*-value, the strongest enrichment was observed in immune system and cytokine related pathways such as "regulation of IFNα signalling", "interferon receptor binding", "cytokine activity", "serine phosphorylation of STAT protein", and "natural

**Table 3. MAGMA results for 18 genes identified using four distinct methods: MAGMA gene-based analysis, positional, eQTL and chromatin interaction mapping.**

| Gene | Status | Chr | Start | End | nSNPs | Z | P-value[a] |
|------|--------|-----|-------|-----|-------|---|---------|
| NEGR1 | confirmed | 1 | 71851623 | 72758417 | 1673 | 5.37 | 3.87E-08 |
| TMEM106B | confirmed | 7 | 12240867 | 12292993 | 272 | 5.40 | 3.30E-08 |
| RP11-145E5.5 | novel | 9 | 21792635 | 22042985 | 412 | 6.91 | 2.40E-12 |
| C9orf53 | novel | 9 | 21957137 | 21977738 | 19 | 6.07 | 6.34E-10 |
| CDKN2A | novel | 9 | 21957751 | 22005300 | 48 | 7.08 | 7.40E-13 |
| CDKN2B | novel | 9 | 21992902 | 22019362 | 37 | 7.85 | 2.05E-15 |
| SPPL3 | novel | 12 | 121190313 | 121352174 | 463 | 5.78 | 3.70E-09 |
| AC079602.1 | novel | 12 | 121397641 | 121420095 | 66 | 6.27 | 1.85E-10 |
| HNF1A | confirmed | 12 | 121406346 | 121450315 | 143 | 5.87 | 2.18E-09 |
| C12orf43 | novel | 12 | 121430225 | 121464305 | 110 | 5.74 | 4.63E-09 |
| OASL | novel | 12 | 121448095 | 121487045 | 118 | 5.75 | 4.42E-09 |
| TTLL6 | novel | 17 | 46829597 | 46904576 | 205 | 5.33 | 4.99E-08 |
| ATP5G1 | novel | 17 | 46960127 | 46983233 | 48 | 5.66 | 7.58E-09 |
| UBE2Z | novel | 17 | 46975731 | 47016418 | 112 | 5.43 | 2.82E-08 |
| SNF8 | novel | 17 | 46996678 | 47032479 | 110 | 5.47 | 2.19E-08 |
| GIP | novel | 17 | 47025916 | 47055958 | 92 | 5.62 | 9.51E-09 |
| IGF2BP1 | novel | 17 | 47064774 | 47143012 | 125 | 5.30 | 5.78E-08 |
| TCF4 | novel | 18 | 52879562 | 53342018 | 668 | 4.75 | 1.03E-06 |

Confirmed status refers to genes that have been associated with all three disorders (i.e., coronary artery disease, type 2 diabetes and depression) in previous studies. Novel status refers to genes that are for the first time being linked to all three disorders. Chr, chromosome; nSNPs, number of single nucleotide polymorphisms; Z, Z-statistic; eQTL, expression quantitative trait loci.

[a]Bonferroni corrected P-value threshold was set at 0.05 / (the number of tested genes) = 2.64e-6.

killer cell activation involved in immune response". For complete results of gene set enrichment analysis see S7 Table. Tissue specificity analysis using 53 specific and 30 general GTEx tissue types demonstrated significant enrichment of upregulated differentially expressed gene sets in kidney ($P_{Bon} < 0.001$; Fig D in S1 Appendix) and kidney cortex tissues ($P_{Bon} = 0.003$; Fig E in S1 Appendix).

## MAGMA gene-set and tissue expression analyses

Out of all 15,485 gene sets tested (in comparison to only prioritised genes above), MAGMA gene set analysis identified five significant gene sets related to processes such as DNA binding, nucleic acid binding, and regulation of respiratory system process (all $P_{Bon} < 0.031$; S8 Table). Lastly, tissue specificity analysis revealed significant gene expression in the cerebellum ($P_{Bon} = 0.040$; Fig F in S1 Appendix) and the pituitary gland ($P_{Bon} = 0.045$; Fig G in S1 Appendix), relative to other tissue types.

## Polygenic risk score analysis

To validate the latent multimorbidity factor, we derived a PRS for multimorbidity in the UKBB cohort ($N = 306,734$). To do this, we first repeated the multivariate GWAS using summary-level data that did not include the UKBB. Results aligned very closely to our discovery GWAS (see results section in S1 Appendix, S4 and S9 Tables for more detail). In UKBB, we observed a dose-response relationship, whereby PRS for multimorbidity was lowest in healthy individuals ($M = -0.037$, $SE = 0.002$), followed by individuals with any one disease ($M = 0.111$, $SE = 0.004$) and any two diseases ($M = 0.379$, $SE = 0.013$). PRS was highest in individuals with

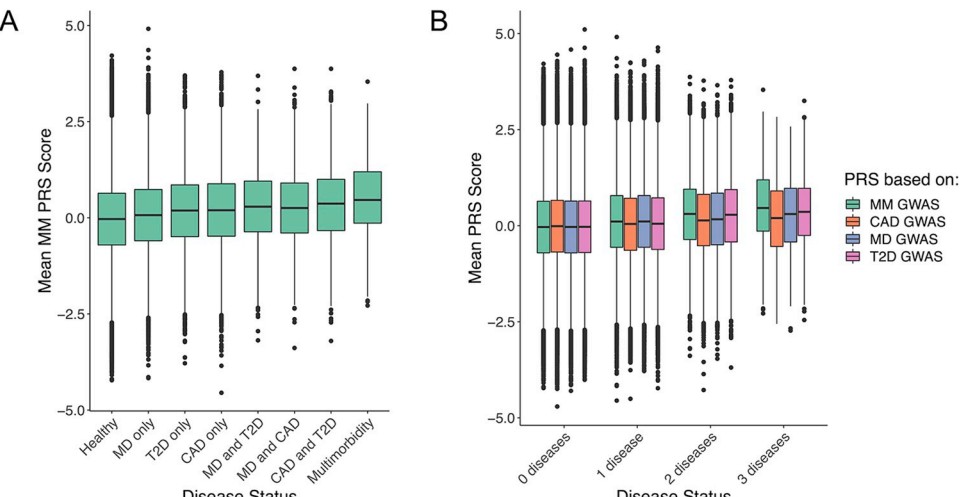

**Fig 3. Out-of-Sample Prediction for Phenotypic Psycho-Cardiometabolic Multimorbidity or Single Diseases using Polygenic Risk Scores.** (A) Multimorbidity polygenic risk score across groups of individuals with no, any one, two or three diseases. (B) Four polygenic risk scores for MD, CAD, T2D, and multimorbidity across groups of individuals with no, any one, two or three diseases. MD, major depression; CAD, coronary artery disease; T2D, type 2 diabetes; MM, multimorbidity; PRS, polygenic risk score.

all three diseases ($M$ = 0.585, $SE$ = 0.046), which aligned well with the common factor structure specified using Genomic SEM (Fig 3; S10 Table) and suggested that findings were not driven solely by the comorbidity between CAD and T2D. Results from multinomial logistic regression revealed that for one standard deviation increase in multimorbidity-PRS, the odds of experiencing multimorbidity (i.e., co-occurrence of CAD, T2D and depression) increased by 91% relative to the healthy group (OR = 1.91, 95% CI = 1.74–2.10). Multimorbidity-PRS was best suited at predicting multimorbidity, rather than any one (OR range = 1.07–1.38) or any two diseases (OR range = 1.46–1.73). Additionally, multimorbidity-PRS outperformed individual PRSs based on CAD, T2D, and depression, especially for the multimorbid group (Table 4). A PRS analysis using only non-heterogeneous SNPs ($Q_{SNP} P \geq 5e-8$) returned almost identical results (S10 Table).

Furthermore, the four PRSs were statistically different in individuals with multimorbidity, $F(3, 1740) = 7.32, P < .001$. Tukey's HSD test indicated that the multimorbidity PRS

**Table 4. Association between four Polygenic Risk Scores and Disease Status in UK Biobank adjusted for Age and Sex using Multinomial Logistic Regression ($N$ = 306,734).**

| Outcome | MM-PRS Adjusted OR (95% CI) | CAD-PRS Adjusted OR (95% CI) | T2D-PRS Adjusted OR (95% CI) | MD-PRS Adjusted OR (95% CI) |
|---|---|---|---|---|
| Healthy | Ref | Ref | Ref | Ref |
| MD only | 1.07 (1.06–1.08) | 1.01 (1.00–1.02) | 1.01 (0.99–1.02) | 1.20 (1.18–1.21) |
| T2D only | 1.36 (1.33–1.38) | 1.13 (1.11–1.15) | 1.42 (1.40–1.45) | 1.07 (1.05–1.10) |
| CAD only | 1.38 (1.35–1.41) | 1.41 (1.38–1.44) | 1.07 (1.05–1.09) | 1.1 (1.07–1.12) |
| MD and T2D | 1.46 (1.40–1.52) | 1.16 (1.12–1.21) | 1.44 (1.38–1.5) | 1.24 (1.19–1.30) |
| MD and CAD | 1.46 (1.39–1.53) | 1.37 (1.31–1.44) | 1.09 (1.04–1.14) | 1.27 (1.21–1.33) |
| CAD and T2D | 1.73 (1.66–1.81) | 1.44 (1.38–1.50) | 1.52 (1.46–1.59) | 1.19 (1.14–1.24) |
| Multimorbidity | 1.91 (1.74–2.10) | 1.53 (1.39–1.68) | 1.53 (1.39–1.68) | 1.38 (1.25–1.51) |

Polygenic risk scores have been standardised. Reference category = healthy. Multimorbidity = CAD + T2D + MD. OR, odds ratio; CI, confidence interval; MM, multimorbidity; CAD, coronary artery disease; T2D, type 2 diabetes; MD, major depression.

($M$ = 0.59, $SE$ = 0.05) was significantly higher in individuals with all three diseases than PRSs for depression ($M$ = 0.29, $SE$ = 0.05, $P_{adj}$ = .001), CAD ($M$ = 0.38, $SE$ = 0.05, $P_{adj}$ = .011), or T2D ($M$ = 0.39, $SE$ = 0.05, $P_{adj}$ = .015; Fig 3).

A sex-stratified analysis using multinomial logistic regression revealed comparable results for males and females. Specifically, relative to the healthy group, for every one standard deviation increase in multimorbidity-PRS, the odds of experiencing multimorbidity increased by 94% (OR = 1.94, 95% CI = 1.73–2.17) for males and 86% (OR = 1.86, 95% CI = 1.57–2.20) for females (S10 and S11 Table).

### Genetic correlations and Mendelian randomization

Results from LD score regression revealed significant genetic correlations with multimorbidity for 17 out of 18 risk factors (all $P_{Bon}$ < 0.002). The strongest genetic correlations were observed for BMI ($r_g$ = 0.60, $SE$ = 0.02, $P_{Bon}$ = 9e-251), body fat percentage ($r_g$ = 0.56, $SE$ = 0.02, $P_{Bon}$ = 3e-174), and C-reactive protein (CRP; $r_g$ = 0.41, $SE$ = 0.04, $P_{Bon}$ = 3e-20). Interestingly, moderate correlations were also observed with insomnia ($r_g$ = 0.36, $SE$ = 0.03, $P_{Bon}$ = 2e-44), neuroticism ($r_g$ = 0.33, $SE$ = 0.02, $P_{Bon}$ = 2e-55), and childhood maltreatment ($r_g$ = 0.33, $SE$ = 0.03, $P_{Bon}$ = 1e-32) (S12 Table).

Mendelian randomization analyses revealed potentially causal associations of BMI, body fat percentage, LDL cholesterol, total cholesterol, fasting insulin, income, insomnia, and childhood maltreatment that survived correction for multiple testing using the Benjamini-Hochberg (BH) false discovery rate. Sensitivity analyses estimates using MR-Egger, simple mode, weighted median and weighted mode methods were generally consistent for all these traits (with a minimum of three out of four sensitivity analyses having significant causal estimates), indicating robustness of our primary results (Fig 4; S13 Table). Evidence for neuroticism, blood pressure traits, and triglycerides was more mixed. We found little or no evidence to support a causal effect of intelligence, worry, sensitivity to environmental stress and adversity, HDL cholesterol, smoking status, and C-reactive protein (Figs H-Y in S1 Appendix; S13 Table). Results from the remaining sensitivity analyses and MRlap are reported in the supplementary material (S1 Appendix, results section and Figs H-Y; S13–S14 Tables).

## Discussion

The present study explored the multivariate genetic architecture of major depression, T2D, and CAD. Assessment of bivariate genetic correlations suggested a shared genetic architecture between all three disorders. The strongest correlation was observed between CAD and T2D ($r_g$ = 0.39), with weaker correlations detected between depression and CAD ($r_{g\,=}$ 0.13) and depression and T2D ($r_g$ = 0.15), suggesting a more distinct genetic basis. This was in line with findings from previous studies which reported genetic correlations of a similar magnitude [44–46].

Akin to the bivariate correlation pattern we observed, results from the factor analysis also revealed that on the genetic level, psycho-cardiometabolic multimorbidity is most representative of CAD and T2D, but less so of depression. Using this factor structure, we identified 11 independent SNPs associated with multimorbidity across nine genomic loci. The direction of effect estimates was concordant across CAD, T2D and depression for seven of the 11 variants, suggesting consistent risk associations with multimorbidity. For the majority of SNPs (n = 7), the largest effects were observed for CAD and T2D. Three SNPs (rs10789340, rs2043539, rs2004910) had comparable effect estimates across the three traits.

Six of the 11 independent SNPs were previously identified as genome-wide significant in the contributing GWAS of CAD [48]. Four of the 11 independent SNPs were also identified as genome-wide significant in the contributing GWAS of T2D [42], and three were identified as

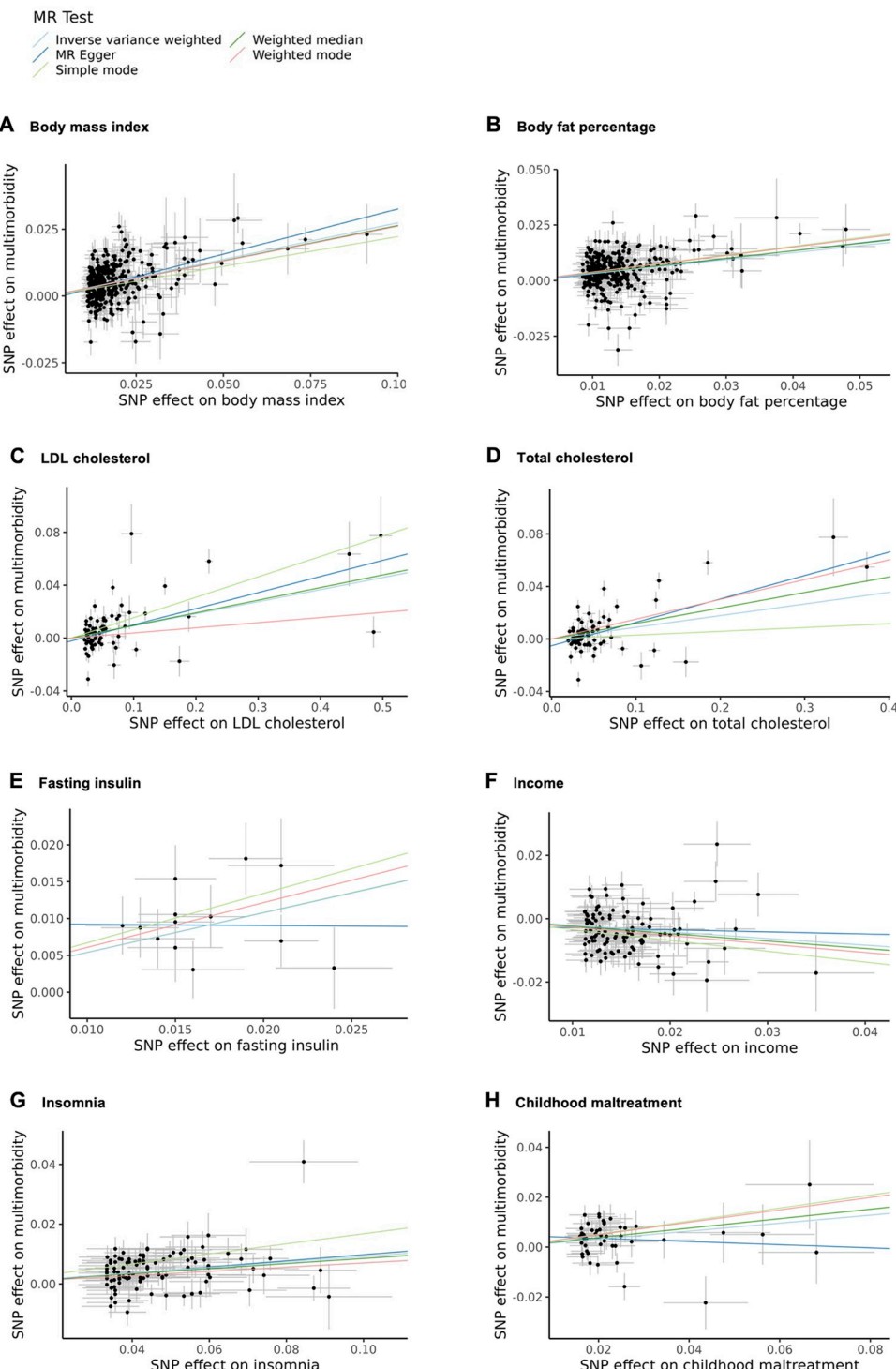

**Fig 4. Scatter plots of Two-Sample Mendelian Randomization results.** Scatter plots showing SNP effects of body mass index (A), body fat percentage (B), LDL cholesterol (C), total cholesterol (D), fasting insulin (E), income (F), insomnia (G), and childhood maltreatment (H) on psycho-cardiometabolic multimorbidity. The slopes represent estimates from the primary (inverse variance weighted) and sensitivity analyses (MR-Egger, simple mode, weighted median, weighted mode). MR, Mendelian randomization; LDL, low-density lipoprotein; SNP, single nucleotide polymorphism.

genome-wide significant in the GWAS of depression [44]. While this was indicative of greater shared genetic architecture underlying CAD and T2D, it could also partially be driven by the smaller effect sizes generally observed for depression-associated SNPs [44]. Weaker associations with depression may also be attributable to greater polygenicity and heterogeneity of depression. Individuals with the same diagnoses of depression may express very different–and even opposing (e.g., increase or decrease in sleep and appetite)–symptom profiles that, in turn, may be related to different pathophysiological mechanisms. For example, previous research has shown that PRSs for higher body mass index, triglycerides [61], C-reactive protein and leptin [62,63] were specifically associated with major depression characterized by atypical symptoms (such as hyperphagia, hypersomnia and weight gain) but not with major depression in general or with other specific subtypes. Thus, lumping different symptom patterns may weaken or dilute genetic associations [64].

Focusing on non-heterogeneous SNPs to minimize genetic signals driven by a single disorder, FUMA prioritised 200 genes putatively associated with multimorbidity, 18 of which were also identified using positional mapping, eQTL mapping, chromatin interaction mapping, and MAGMA gene-based analysis, thereby providing the most consistent support for these genes. Three of these genes (*NEGR1*, *TMEM106B*, *HNF1A*) have been linked to each of the three diseases (CAD, T2D and depression) in previous studies [44,65–74], supporting their probable involvement in psycho-cardiometabolic multimorbidity. The remaining 15 were completely novel (e.g., *SNF8* and *AC079602.1*) or had only been linked to one or two diseases. For example, while *SPPL3* and *TCF4* have been associated with T2D [42,67] and depression [72,75,76] (as well as various other psychiatric traits [77,78]), an association with CAD has not yet been reported. However, *SPPL3* and *TCF4* have been linked to CAD risk factors such as cholesterol levels [79,80] and CRP [79,81,82], suggesting a potential role in multimorbidity. Similarly, *UBE2Z*, *GIP* and *IGF2BP1* have been linked to CAD [70,83,84], T2D [66,67,85], and depression-related traits such as insomnia, BMI, educational attainment, CRP levels, platelet count and smoking [82,86–92]. As such, even though a direct association with depression has not yet been established, their relevance in psycho-cardiometabolic multimorbidity seems biologically plausible.

A similar pattern was observed for the other genes, whereby the majority tagged common risk factors for CAD, T2D and depression, such as adiposity related traits (BMI, body fat percentage, waist/hip circumference), inflammatory markers (CRP, interleukin-6, interleukin-5), lipids (low- and high-density lipoprotein levels), platelet traits (platelet count, plateletcrit), and N-glycan levels [42,70,79,87,93–100]. Previous knowledge for three of the identified genes (*RP11-145E5.5*, *SNF8*, *AC079602.1*) was weak, suggesting potentially novel targets for follow-up in relation to multimorbidity.

The prioritised genes showed an enrichment in immune and cytokine related pathways, which are involved in the regulation of immune and inflammatory responses–both of which have been implicated in the pathophysiology of CAD, T2D and depression. For example, interferons and cytokines play a central role in the innate immune system and in the initiation of inflammatory cascades [101,102]. Experimental and longitudinal studies suggest that for a significant subset of patients, immune system dysfunction in general and inflammation in particular may be causally implicated in the development of depression [103–106]. A recent study showed that higher interleukin 6 activity is potentially causal especially for specific symptoms of depression, such as sleep problems or fatigue [107]. Similarly, chronic inflammation has been identified as a feature of CAD, promoting the growth of plaques in the arteries and worsening clinical outcomes, irrespective of serum lipid levels [108,109]. Innate and adaptive immunity, together with low-grade inflammation have also been recognised as important aetiological factors in the pathogenesis of insulin resistance and T2D [110]. Hence, the

implicated genes and biological processes reflect biological plausibility for shared genetic aetiology between CAD, T2D and depression. On the other hand, it is also possible that an inflammatory response is the downstream effect of these diseases.

Tissue specificity analysis indicated increased gene expression relative to other tissue types in the cerebellum and the pituitary gland. This is of interest as the structure and function of these regions seems to be altered in depression. For example, individuals with depression tend to have an overactive hypothalamic-pituitary-adrenal axis (our main stress response system), leading to increased cortisol levels and suppressed immune responses [20]. Similarly, with regard to the cerebellum, important cerebellar alterations have been identified in patients with depression [111], impacting emotion regulation ability [112]. Despite the limited contribution of depression to the latent multimorbidity factor, the involvement of these two regions provides reassurance that depression is captured in our analysis. This is further supported by genetic correlation and MR results between a number of depression-related risk factors, such as insomnia, childhood maltreatment and adiposity traits.

Overall, while we observed weak-to-strong genetic correlations between 17 risk factors and multimorbidity, only eight of these associations (BMI, body fat percentage, LDL cholesterol, total cholesterol, fasting insulin, income, insomnia, and childhood maltreatment) demonstrated consistent estimates across most MR analyses, suggesting potentially causal effects.

The findings of the present study should be interpreted in light of the following limitations. First, we only considered common genetic variants, but it is also possible that multimorbidity is driven by rare variants with minor allele frequencies below 1%. Second, the contributing GWAS by Nikpay et al.[48] included mixed ancestry individuals (23%), which is cautioned against when using Genomic SEM. However, as the LD score intercepts for CAD and multimorbidity GWASs were close to 1 (0.88 and 0.99, respectively), suggests that our results are unlikely to be biased due to ancestry issues. Third, although we removed SNPs with strong evidence for heterogeneity ($Q_{SNP}$ $P < 5e-8$ and directionally discordant univariate effect estimates), there were still many variants left in the analyses with suggestive evidence for heterogeneity. This means that our downstream analyses may be biased towards pathways related to any one or two constituting diseases. Therefore, when using multimorbidity summary statistics in future studies, it may be appropriate to apply an even more stringent heterogeneity threshold (e.g., $P < 5e-6$), depending on the nature of the investigation.

Fourth, multimorbidity was defined by the common factor structure we specified using Genomic SEM, where the latent variable accounted for the largest proportion of variance in T2D and CAD, with a smaller amount of variance explained in depression. This had implications for the identification of genetic variants and the prioritisation of genes, which were based on the latent multimorbidity factor and were therefore capturing depression to a lesser extent. Genetic variants identified based on such a factor structure may put into question the interpretability of current results, as SNPs for multimorbidity may simply reflect the prespecified factor structure (i.e., an effect of the GWASs used) rather than a robust finding. However, considering that (1) the mean PRS was larger in individuals with all three diseases compared to those with any one or two diseases and (2) a multimorbidity PRS (as opposed to PRSs for single diseases) was most strongly associated with multimorbidity phenotype, suggests that the present study detected putative pleiotropic variants that influence CAD, T2D and depression.

In summary, the present study investigated the shared genetic architecture across CAD, T2D and depression and performed a multivariate GWAS of psycho-cardiometabolic multimorbidity. The analysis identified 11 independent SNPs associated with multimorbidity and 18 putative multimorbidity-associated genes. Three of these genes had already been linked to each of the three diseases in previous studies and 15 were novel or had only been linked to one or two diseases. The prioritised genes were enriched in immune and inflammatory pathways,

elucidating putative biological mechanisms underlying psycho-cardiometabolic multimorbidity. Considering that susceptibility to CAD, T2D and depression is also influenced by environmental factors [113–116], future studies should explore multimorbidity in the context of gene-environment correlations and interactions. Lastly, to decipher the role of depression heterogeneity, similar analyses could be performed using subgroups of individuals characterized by different depression profiles (e.g., atypical symptoms, inflammation).

Overall, our findings advance our understanding of genetic associations related to multimorbidity and provide avenues for future research.

## Web resources

Genomic SEM: https://github.com/GenomicSEM/GenomicSEM/wiki.

LDSC package in Python: https://github.com/bulik/ldsc.

FUMA GWAS online platform: https://fuma.ctglab.nl.

PRSice-2: https://choishingwan.github.io/PRSice/step_by_step/.

LocusZoom: http://locuszoom.org/.

## Supporting information

**S1 Table. Population characteristics of the input genome-wide association meta-analyses.**
(XLSX)

**S2 Table. LD score regression results for univariate (input) and multivariate genome-wide association studies.**
(XLSX)

**S3 Table. Genomic risk loci from the multivariate GWAS of psycho-cardiometabolic multimorbidity.**
(XLSX)

**S4 Table. Genomic SEM output for independent significant SNPs from the multivariate GWAS of psycho-cardiometabolic multimorbidity.**
(XLSX)

**S5 Table. Univariate coefficients and their standard errors scaled relative to unit-variance scaled phenotypes.**
(XLSX)

**S6 Table. A list of prioritized genes by functional mapping in the discovery GWAS of psycho-cardiometabolic multimorbidity.**
(XLSX)

**S7 Table. Results of gene set enrichment analysis.**
(XLSX)

**S8 Table. MAGMA gene-set analysis containing one P-value per gene set.**
(XLSX)

**S9 Table. A list of prioritized genes by functional mapping in the multivariate GWAS of psycho-cardiometabolic multimorbidity (version without UK Biobank).**
(XLSX)

**S10 Table. Proportion of multimorbid individuals in UK Biobank and their polygenic risk scores calculated using PRSice-2.**
(XLSX)

**S11 Table. Sex-stratified results for the association between multimorbidity polygenic risk score and disease status in UK Biobank.**
(XLSX)

**S12 Table. Results of LD Score regression analyses between psycho-cardiometabolic multimorbidity and various risk factors.**
(XLSX)

**S13 Table. Results from two-sample Mendelian randomization using inverse variance weighted method as the primary analysis and MR-Egger, simple mode, weighted median and weighted mode methods as sensitivity analyses.**
(XLSX)

**S14 Table. Results for Mendelian randomization Steiger test of directionality.**
(XLSX)

**S1 Appendix. Supplementary Appendix.**
(PDF)

**S1 Text. FUMA SNP2GENE parameters.**
(DOCX)

**S2 Text. FUMA GENE2FUNC parameters.**
(DOCX)

## Acknowledgments

We are grateful to the UK Biobank and all its voluntary participants. We also thank all 23andMe research participants who made this study possible.

## Author Contributions

**Conceptualization:** Vilte Baltramonaityte, Jean-Baptiste Pingault, Charlotte A. M. Cecil, Janine Felix, Sylvain Sebert, Yuri Milaneschi, Esther Walton.

**Formal analysis:** Vilte Baltramonaityte.

**Methodology:** Vilte Baltramonaityte, Jean-Baptiste Pingault, Yuri Milaneschi, Esther Walton.

**Supervision:** Sylvain Sebert, Esther Walton.

**Writing – original draft:** Vilte Baltramonaityte.

**Writing – review & editing:** Vilte Baltramonaityte, Jean-Baptiste Pingault, Charlotte A. M. Cecil, Priyanka Choudhary, Marjo-Riitta Järvelin, Brenda W. J. H. Penninx, Janine Felix, Sylvain Sebert, Yuri Milaneschi, Esther Walton.

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
