## [Decision Letter · Decision Letter 0]

17 Jan 2023

Dear Dr Walton,

Thank you very much for submitting your Research Article entitled 'A multivariate genome-wide association study of psycho-cardiometabolic multimorbidity' to PLOS Genetics.

The manuscript was fully evaluated at the editorial level and by independent peer reviewers. The reviewers appreciated the attention to an important problem, but raised some substantial concerns about the current manuscript. Based on the reviews, we will not be able to accept this version of the manuscript, but we would be willing to review a much-revised version. We cannot, of course, promise publication at that time.

If you decide to revise the manuscript for further consideration at PLOS Genetics, please aim to resubmit within the next 60 days, unless it will take extra time to address the concerns of the reviewers, in which case we would appreciate an expected resubmission date by email to plosgenetics@plos.org.

We are sorry that we cannot be more positive about your manuscript at this stage. Please do not hesitate to contact us if you have any concerns or questions.

Yours sincerely,

Heather J Cordell

Academic Editor

PLOS Genetics

Scott Williams

Section Editor

PLOS Genetics

Reviewer's Responses to Questions

**Comments to the Authors:**

Reviewer #1: This paper is an interesting advance to the rapidly expanding field of “multimorbidity” – usually defined as the study of patients with more than one long term condition. The paper uses a genomic SEM approach to define a “latent phenotype” that is based on the overlap between three common conditions – type 2 diabetes, heart disease and depression. They then perform a GWAS of this latent phenotype and identify associated variants. Critically , the authors take the analysis a step further and use a heterogeneity statistic to assess whether or not variants associated with their latent multimorbidity phenotype are likely to be genuinely contributing to it independently of main effects on one of the traits. This is an important methodological advance for the field, but the approach looks like it has not resulted in a large number of new, genuine “multi-morbidity” signals. The well known signals on chromosome 9 that are involved in diabetes and heart disease look convincing, but most of the variants show some sign that they are mainly driven by one of the 3 traits – according to the heterogeneity stats in the supplementary tables. The data in table 3 is more promising – a multi morbidity polygenic risk score is increasing the risk of having all 3 conditions more than having just two of them. This result suggests the variants are capturing some of the depression component not just the CAD and T2D components to the latent phenotype.

Major points

1. The CAD data used needs updating. The Nikpay et al. paper is I believe the largest GWAS of CAD without including UK Biobank, and includes 60,000 cases: PMID: 26343387 . This will increase power substantially from the Schunkert et al. data using 22,000 cases – especially as the statistical confidence of some of the genetic correlations looks relatively weak.

2. An r2 of 0.6 is a very high threshold for declaring independence of signals. In these sample sizes variant- associations where the variant has an r2 of 0.5 with another variant could easily be a “shadow” of another variant. I suggest present results from r2 < 0.01. This would only affect the number of signals in the chr 9 locus and may aide interpretation of that region.

3. A novel aspect of the paper is the use of the heterogeneity statistic. This Q stat helps reveal which variants could be primarily driven by single disease associations rather than be genuine multi morbidity signals. This is a key aspect of the paper and I think the Q stats should be added to table 2.

Minor points:

1. Although we cannot assume the closest gene is causal, because many common GWAS signals are known by local gene names, such as CDKN2 in the Chr 9 locus, it would be helpful to add closest/nearest gene labels to the key results tables.

2. Suggest do not use familiar phrases such as “to get at” in the context of “to get at the shared variance” etc.

3. Line 346 – the standard definition of multi morbidity is living with 2 or more conditions, but the text implies all 3 ?

Reviewer #2: Baltramonaityte et al have performed a genome-wide association study of psycho-cardiometabolic multimorbidity using genomic SEM that is well-written and easy to follow. I have some comments detailed below:

1. BMI is known to be causally related to both depression, CAD and T2D in mendelian randomization studies, yet not mentioned in the paper as a possible reason for this co-morbidity. I find this a bit odd and wonder about the rational for not including such a major determinant in either the introduction or the analysis. At the very least the reasoning behind this decision should be discussed.

2. I think it would be useful if the authors provided their genomic SEM script in the supplementary.

3. Reference UKBB phenotype information in supplementary text in main text.

4. The manuscript could potentially be made more interesting by trying to elude the causal effect of this psycho-cardiometabolic multimorbidity, e.g. mendelian randomization to find exposures causally associated with the multimorbidity or run LD score regression with the multimorbidity summary statistics to test which other traits it has a genetic correlation with.

Reviewer #3: The authors used genome wide association studies of coronary heart disease , T2D and depression to investigate the shared genetic factors between these three conditions.

The introduction is unconvincing in why the authors decided to include T2D and link this to depression is unclear, biological mechanisms were not explained.

The authors did not use the latest CKD GWAS. They used one published in 2011 so quite outdated and with a much fewer number of individuals and significant snps. Without the use of the latest GWAS the publication of the paper will be already outdated so I suggest they re-do it.

I would expect that the sensitivity analysis would only include cohorts where depression diagnosis was properly validated using structured clinical questionnaire, and not including only those with depressive symptoms. However this important point for this condition in particular was not discussed.

The authors did not investigate any cardiovascular, diabetes or depression risk factors. The authors did not investigate heterogeneity. There were no functional analysis of the shared loci. The authors did not separate the findings by gender which would be important in this context.

The paper is in a way very similar idea to the already published article in PLOS Genetics 2022 Shared genetic loci between depression and cardio metabolic traits, but of much lower quality.

**Have all data underlying the figures and results presented in the manuscript been provided?**

Reviewer #1: Yes

Reviewer #2: Yes

Reviewer #3: Yes

PLOS authors have the option to publish the peer review history of their article (what does this mean?). If published, this will include your full peer review and any attached files.

Reviewer #1: **Yes: **Timothy Frayling

Reviewer #2: No

Reviewer #3: No

---

## [Decision Letter · Decision Letter 1]

5 May 2023

Dear Dr Walton,

Thank you very much for submitting your Research Article entitled 'A multivariate genome-wide association study of psycho-cardiometabolic multimorbidity' to PLOS Genetics.

The manuscript was fully evaluated at the editorial level and by independent peer reviewers. The reviewers appreciated the improvements made, but one reviewer made some good suggestions that we ask you address in a further revised manuscript.

We therefore ask you to modify the manuscript according to the review recommendations. Your revisions should address the specific points made reviewer 1.

Yours sincerely,

Heather J Cordell

Academic Editor

PLOS Genetics

Scott Williams

Section Editor

PLOS Genetics

Reviewer's Responses to Questions

**Comments to the Authors:**

Reviewer #1: Many thanks for doing such a thorough job answering my queries. My only final comment is that the discussion limitation section ideally needs to be a little more circumspect about how well the latent factor is genuinely representing MM or just the individual conditions. Ten of the 11 variants in the main table reach p<0.05 and most are p<5x10-8 for their heterogeneity values, indicating they are primarily driven by one trait. the use of 5x10-8 as the QSNP thershold is fine but could still mean you have a lot of enrichment for heterogeneous variants, making all the downstream analysis biased towards finding pathways affecting one trait. A quick QQ plot of the QSNPs could englighten this issue ?

Reviewer #2: I have no further comments.

Reviewer #3: I am satisfied with the changes made

**Have all data underlying the figures and results presented in the manuscript been provided?**

Reviewer #1: Yes

Reviewer #2: Yes

Reviewer #3: Yes

PLOS authors have the option to publish the peer review history of their article (what does this mean?). If published, this will include your full peer review and any attached files.

Reviewer #1: **Yes: **Timothy M Frayling

Reviewer #2: No

Reviewer #3: No

---

## [Decision Letter · Decision Letter 2]

12 Jun 2023

Dear Dr Walton,

We are pleased to inform you that your manuscript entitled "A multivariate genome-wide association study of psycho-cardiometabolic multimorbidity" has been editorially accepted for publication in PLOS Genetics. Congratulations!

Yours sincerely,

Heather J Cordell

Academic Editor

PLOS Genetics

Scott Williams

Section Editor

PLOS Genetics

Comments from the reviewers (if applicable):

Reviewer's Responses to Questions

**Comments to the Authors:**

Reviewer #1: thanks for doing such a thorough job at responding to my final query and making the code available.

**Have all data underlying the figures and results presented in the manuscript been provided?**

Reviewer #1: Yes

PLOS authors have the option to publish the peer review history of their article (what does this mean?). If published, this will include your full peer review and any attached files.

Reviewer #1: **Yes: **Timothy Frayling

**Data Deposition**

http://datadryad.org/submit?journalID=pgenetics&manu=PGENETICS-D-22-01265R2

**Press Queries**

---

## [Editor Report · Acceptance letter]

26 Jun 2023

PGENETICS-D-22-01265R2 

A multivariate genome-wide association study of psycho-cardiometabolic multimorbidity 

Dear Dr Walton, 

We are pleased to inform you that your manuscript entitled "A multivariate genome-wide association study of psycho-cardiometabolic multimorbidity" has been formally accepted for publication in PLOS Genetics! Your manuscript is now with our production department and you will be notified of the publication date in due course.

With kind regards,

Zsofia Freund

PLOS Genetics

On behalf of:
